# Effects of a Carbon Nanotube Additive on the Corrosion-Resistance and Heat-Dissipation Properties of Plasma Electrolytic Oxidation on AZ31 Magnesium Alloy

**DOI:** 10.3390/ma11122438

**Published:** 2018-12-02

**Authors:** Myungwon Hwang, Wonsub Chung

**Affiliations:** Department of Materials Science and Engineering, Pusan National University, Busan 609-735, Korea; mwhwang@pusan.ac.kr

**Keywords:** magnesium alloy, plasma electrolytic oxidation, carbon nanotube, corrosion resistance, emissivity

## Abstract

Plasma electrolytic oxidation (PEO) coating was obtained on AZ31 Mg alloy using a direct current in a sodium silicate-based electrolyte with and without a carbon nanotube (CNT) additive. The surface morphology and phase composition of the PEO coatings were investigated through field emission scanning electron microscopy (SEM), X-ray diffraction (XRD), and X-ray photoelectron spectroscopy (XPS). The corrosion-resistance properties of the PEO coatings were evaluated using potentiodynamic polarization measurements and electrochemical impedance spectroscopy (EIS) in a 3.5 wt.% NaCl solution. Furthermore, the heat-dissipation property was evaluated by a heat-flux measurement setup using a modified steady-state method and Fourier transform infrared spectroscopy (FT-IR). The results demonstrate that, by increasing the concentration of CNT additive in the electrolyte, the micropores and cracks of the PEO coatings are greatly decreased. In addition, the anticorrosion performance of the PEO coatings that incorporated CNT for the protection of the Mg substrate was improved. Finally, the coating’s heat-dissipation property was improved by the incorporation of CNT with high thermal conductivity and high thermal emissivity.

## 1. Introduction

Magnesium alloys have been widely used in the automobile, aerospace, and electronics industries due to their low density, high specific strength, specific stiffness, castability, and machinability [1,2]. However, they also have important constraints due to their poor corrosion resistance, which is caused by low active electrochemical potential. To overcome this limitation, surface-treatment techniques, such as physical vapor deposition (PVD), thermal spraying, and anodizing, have been developed [3,4,5].

Plasma electrolytic oxidation (PEO), a typical surface-treatment technology based on conventional anodization, is environmentally friendly and economical [6,7,8]. Ceramic-like oxide coating is formed on the magnesium substrate to improve corrosion and abrasion resistance. Since PEO proceeds above the breakdown voltage, a microarc occurs on the surface of the oxide film, and the oxide film is locally melted and mixed with electrolyte ions. Using these process features, various additives have been used to improve the performance of the coating. Composite PEO can be divided into two types: it either uses a water-soluble additive or an insoluble additive [9]. The former is used in the form of an aqueous solution, mixed with basic electrolytes, and the latter is formed by adding insoluble micro- and nanosized particles to the electrolyte to form a composite PEO coating. 

Heat transfer is classified into three main mechanisms, thermal conduction, thermal convection, and thermal radiation [10]. A heatsink cooled by natural convection uses the conduction and radiation mechanisms. An oxide coating is generally applied to improve the heat-dissipation capabilities of the heatsink [11,12,13]. Many studies have been carried out to optimize shape and material to improve thermal conductivity [14,15,16]. However, heat dissipation using radiation only uses conventional anodizing, and research on the improvement of heat-dissipation performance through high emissivity is insufficient. The heatsink with high emissivity is advantageous for the miniaturization and weight reduction of the product by ensuring sufficient heat-dissipation capacity with a small volume. It is also advantageous to reduce cost by using few raw materials.

In this paper, a carbon nanotube (CNT) with excellent emissivity and thermal conductivity was incorporated into the oxide film to produce coatings with excellent heat-dissipation capabilities. Radiation heat dissipation can be expressed with a modified Stefan–Boltzmann law as [17,18]: E = ε × σ × (T_h_^4^ − T_c_^4^) × A(1)where E, ε, σ, T_h_, T_c_, and A are radiative dissipated heat energy, emissivity (0 ≤ ε ≤ 1), Stefan–Boltzmann constant (5.6703 × 10^−8^ W/m^2^·K^4^), hot-body absolute temperature (K), cold-surroundings absolute temperature (K), and the surface area of the object (m^2^), respectively. In other words, emissivity is a major factor in the heat transfer using the radiation mechanism. When emissivity is close to 1, as in the case of a black body, the heat-dissipation property due to radiation increases and can be applied to a heat sink.

The emissivity of MgO is 0.68, which shows somewhat low emissivity [19]. On the other hand, CNT shows 0.98 emissivity, close to that of a black body [20]. Therefore, if CNT is embedded in oxide film with composite PEO, it can allow the oxide film to have high corrosion resistance and emissivity. 

## 2. Experimental Procedure 

A plate, sized 40 × 80 × 2 mm, of AZ31 Mg alloy was used for all experiments. The samples were polished with SiC paper to approximately 1200, washed with distilled water at room temperature, and ultrasonically cleaned in ethanol. The area except for the area of 40 × 40 mm where the PEO treatment is to be performed were masked with a heat-resistant insulating tape.

A schematic view of the electrolytic cell is depicted in Figure 1. A DC supply of 2.4 kW (600 V-4 A) was used, and 304 stainless steel was used as a counter-electrode. The composition of the electrolytic solution is shown in Table 1. Each electrolyte solution was stirred at 400 rpm with a magnetic stirrer for 60 min before the experiment. At the same time, the CNT (Multiwalled CNT, Carbon Nanomaterial Technology, Pohang-si, Korea) was uniformly dispersed physically by a horn-type ultrasonicator (ULH-700S, Sibata, Soka-shi, Japan). The temperature of the electrolyte was maintained at 20 ± 2 °C through the chiller (DA-1000B, Daeil, Busan-si, Korea). The PEO process maintained 10 amperes per square decimeter (ASD) through the electrostatic method. The voltage change over time was measured by a multimeter (FLUKE 289, FLUKE, Everett, WA, USA) throughout the PEO process. After the PEO process, the sample was washed with distilled water.

The surface morphology and cross-sectional views of the coated samples were observed using a field-emission scanning electron microscope (FE-SEM, Mira 3, Tescan, Brno, Czech Republic). Qualitative phase analysis was performed with X-ray diffraction (XRD, Ultima IV, Rigaku, Akishima-shi, Japan) using a Cu target in the range of 20 to 80 degrees. The chemical state was analyzed using X-ray photoelectron spectroscopy (XPS, Sigma Probe, Thermo Fisher Scientific, Waltham, MA, USA).

All electrochemical experiments were carried out using a 250 mL flat cell with three electrode structures. Platinum sheets of 30 × 30 mm^2^ and saturated calomel electrodes (SCE) were used as counter-electrodes and reference electrodes, respectively. In the 3.5 wt.% NaCl solution, the working electrode was exposed in an area of 1 cm^2^, and the experiment proceeded at room temperature.

The potentiodynamic polarization experiment was performed using Versastat4 (Ametek, France). Before carrying out the potentiodynamic experiment, it was held for about 10 min to stabilize the open-circuit potential (OCP). The polarization curves were obtained in the range of −1.2 V/SCE to 1 V/SCE, and the experiment was performed at a scan rate of 1.67 mV/s. Electrochemical Impedance Spectroscopy (EIS) experiments were performed using IM6 (ZAHNER-Electrick, Kronach, Germany). The EIS experiments were carried out after stabilization in the electrolytic solution for about 30 min. The test was performed in the frequency range of 100 kHz to 0.01 Hz under the condition of applying an AC voltage of 10 mV to the OCP. 

The heat flux was measured with an apparatus used in a previous study, and the heat-flux instrument is shown in Appendix A [21]. Moreover, Fourier transform infrared spectroscopy (FT-IR, Nicolet 360, Thermo Fisher Scientific, Waltham, MA, USA) was used to measure the emissivity for thermal-radiation performance at 100 °C. 

## 3. Results and Discussion

### 3.1. Plasma Electrolytic Oxidation Process

As described in Table 1, the pH was constant regardless of the amount of CNT added, and the electrical conductivity of the electrolyte increased from 13.7 to 18.1 mS/cm as high-conductivity CNT was added. Dispersion of CNT from the electrolyte was affected by zeta potential. The isoelectric point of CNT was 4.5 pH [22]. The zeta potential increased with increasing pH [23]. An increase in the absolute value of the zeta potential causes electrostatic repulsion between the particles and improves dispersion stability. The zeta potential of CNT dispersed with the silicate-based electrolyte was negatively charged at −66.8 mV, and stable dispersion was achieved. Therefore, negatively charged CNT was attracted to the positive electrode and incorporated into the film during the PEO process.

### 3.2. Morphology and Structure of PEO Coatings

Figure 2 shows voltage change over time during the PEO process along with the concentration of CNT in the electrolyte. The change in voltage over time can be explained in three stages [24]. The first stage is when voltage increases rapidly. At this time, a transparent passive film acting as an electrical insulator grows rapidly. The second stage is when the slope of the voltage drops rapidly, and the breakdown voltage appears. Microdischarge starts to occur after the appearance of the breakdown voltage. The third stage is when a large-scale arc occurs with a stable slope. Regardless of the amount of added CNT, the voltage-response behavior remained similar over time in the first stage. However, in the second and third stages, as the CNT amount was increased, breakdown voltage lowered rapidly, and the voltage reaching equilibrium was also lower. Since breakdown voltage is influenced by the electrolyte, the electrical conductivity of the electrolyte is increased by adding CNT, and breakdown voltage is decreased [25,26]. In addition, it is presumed that voltage fluctuation is formed by electrophoresis and physical stirring of CNT.

Figure 3 is an optical photograph of the sample according to the amount of CNT added. Samples without addition of CNT are white and gradually darken with addition of CNT. When the CNT concentration of the electrolyte is 10 g/L, the surface of the sample is black. It can be seen that the color of the PEO coating changes depending on the amount of CNT incorporation. 

The surface morphologies of coatings formed on AZ31 Mg alloy specimens at different concentrations of CNT are shown in Figure 4. The PEO coating without CNT shows a typical porous structure with some bulges on the surface [27]. Surface pores are formed by the solidifying oxide film melted by the microarcs including spherical gas bubbles [28]. Microcracks in the PEO coatings occur due to thermal stresses during coating formation as a result of the melting and solidification of ceramic compounds, such as MgO and Mg_2_SiO_4_ [29,30]. Figure 4d shows the surface morphology of the PEO coating with 10 g/L CNT additive. Due to the high electrical conductivity of the CNTs in the electrolyte, breakdown voltage and working voltage are significantly reduced, and a small microarc is generated. As a result, the porosity and size of the microcracks decreased. It was also seen that the thermal stress was reduced by the weak discharge, and the cracks in the coating layer were reduced. Accordingly, the structure of the oxide film became much finer. The high-magnification SEM micrograph of inset Figure 4d indicates the incorporation of CNT around the pores.

Figure 5 shows the SEM cross-sectional images of the PEO coatings processed in electrolytes containing different concentration of CNT additive. The cross-sectional morphology of the PEO coating consists of three parts: outer porous layer, pore band, and inner dense layer at the substrate/coating interface [31]. Pore bands can be formed between the outer and inner layer of PEO coating using the direct current mode [32]. The average thickness of the coating layer decreased to 19.3, 14.2, 12.5, and 8.0 μm as the addition amount of CNT increased to 0, 2.5, 5, and 10 g/L, respectively. It seems that the thickness of the coating layer was thinned by the low energy input due to the low operating voltage during the PEO process. In addition, the film that is formed by the small microarc generated by the addition of CNT appears to have a denser structure. In general, the corrosion of the PEO coating occurs when corrosive substances reach the Mg substrate through pores or microcracks and form a large amounts of Mg(OH)_2_. As the volume of Mg(OH)_2_ is larger than that of MgO, the interface causes cracking and promotes corrosion [33]. The insoluble additive in the PEO coating exhibits a self-sealing effect for the micropores and cracks in the coating [34]. CNT present in pores and microcracks is likely to inhibit the evolution of corrosive ions.

### 3.3. Phase Analysis and Chemical State of PEO Coatings

X-ray diffraction patterns of PEO coating samples with and without CNT additives and uncoated AZ31 Mg alloy substrates are illustrated in Figure 6. Since an X-ray can easily penetrate the porous oxide layer, a strong diffraction peak of the Mg substrate was detected in all samples. The main phases of oxide films were composed of MgO and Mg_2_SiO_4_ phases in the silicate-based electrolyte [24,35]. MgO forms on the surface of magnesium substrates under high temperatures and a strong electric field, as shown by the following anodic reaction [36]. Mg → Mg^2+^ + 2e^−^(2)
H_2_O → O_2_ + 4H^+^ + 4 e^−^(3)
Mg^2+^ +2OH^−^ → Mg(OH)_2_(4)
Mg(OH)_2_ →MgO + H_2_O(5)

Mg_2_SiO_4_ phases in a silicate-based electrolyte are produced according to the following equations:Mg^2+^ + 2SiO_3_^2−^ → Mg_2_SiO_4_ + SiO_2_(6)

In addition, molten MgO reacts with molten SiO_2_ to form Mg_2_SiO_4_ in the PEO ambient. SiO_3_^2−^ +2H^+^ → SiO_2_ + H_2_O(7)
2SiO_3_^2−^ → O_2_ + 2SiO_2_ + 4e^−^(8)
SiO_2_ + 2MgO → Mg_2_SiO_4_ + SiO_2_(9)

There is no CNT signal detected in the XRD pattern, due to the small amount of CNT incorporation. Furthermore, the CNT addition seems to have a negligible effect on the phase composition of the PEO coating.

Figure 7 show the XPS spectra of the PEO coatings formed in an electrolyte with and without 10 g/L CNT additive. The XPS spectra detected Mg, O, Si, and C peaks. The C 1s peak of the CNT-free coating seems to be hydrocarbon contamination inside the XPS vacuum chamber [37]. In the case of the CNT-added coating, it is shown that the C 1s peak is significantly stronger than the PEO coating without CNT, and this increase in peak intensity means that CNT can be incorporated into the oxide film through the PEO process. Figure 8 represents the high-resolution spectra of the C 1s peak. Curve fitting was performed after a Shirley background subtraction by the mixed Gaussian–Lorentzian function. The C 1s signals on the pristine CNT and the incorporated CNT in the PEO coating at 284.4–284.6, 284.9–285.3, 286.0–286.4, and 287.3–287.7 eV corresponded to sp^2^ C=C, sp^3^ C–C, C–O, and C=O chemical bonding, respectively [38]. It was confirmed that CNT oxidation occurred during the PEO process due to the large increase of C–O and C=O signals compared to the signal of pristine CNT. The surface of the CNT was damaged by the excessive oxidation, and the length thereof was cut off, causing defects in the CNT that may lead to the deterioration of the CNT properties [39].

### 3.4. Corrosion Behavior of PEO Coatings

The potentiodynamic polarization curve of the AZ31 Mg alloy and the PEO coatings with different amounts of CNT additive in the electrolyte are shown in Figure 9. Corrosion potential (E_corr_), corrosion density (i_corr_), and the Tafel slope were used to characterize the corrosion resistance of the samples. The high corrosion potential and low corrosion current density indicate the improving corrosion resistance of the oxide film. The electrochemical parameters derived by the Tafel extrapolation technique are presented in Table 2. From Table 2, we can see that AZ31 Mg alloy has poor corrosion resistance due to high corrosion current density (−1.481 V) and low corrosion potential (1.12 × 10^−5^ A/cm^2^). On the other hand, as CNT concentration of the electrolyte increased in the PEO process, corrosion resistance was highly improved. In particular, for samples with 10 g/L CNT additive added to the electrolyte, the value of the corrosion current density was 4.80 × 10^−7^ A/cm^2^, a decrease of about two orders of magnitude relative to the uncoated AZ31 Mg alloy. These results demonstrate that the incorporation of a CNT additive into the oxide film improves the corrosion-resistance properties of the PEO coating. 

In order to understand more details about the corrosion behavior of the PEO coating in 3.5 wt.% NaCl, EIS measurements were conducted. Figure 10 represents the Nyquist and Bode plots of uncoated AZ31 Mg substrate and PEO-coated samples with different amounts of CNT added into the electrolyte. As shown in Figure 10a, the Nyquist plot of the uncoated AZ31 Mg alloy shows a capacitive loop in the high-frequency range and an inductive loop in the low-frequency range. The Nyquist plot of the PEO-coated magnesium alloy shows two partially overlapped capacitive loops: a small one at lower frequencies and another at intermediate frequencies. Such behavior indicates the presence of two time constants. As the concentration of CNT increases, the radii of the two capacitive loops increase. As per the Bode plots in Figure 10b, the impedance value of the uncoated AZ31 Mg alloy was about 10^2^ Ω⋅cm^2^ in the low-frequency region, and the impedance of the PEO coating in the same region was measured at about 10^4^ Ω⋅cm^2^, and increased as high as 100 times. In particular, the impedance of the PEO-coating layer with the concentration of 10 g/L CNT additive in the electrolyte was 10^5^ Ω⋅cm^2^, which is the highest value. The applied equivalent circuit is a widely applied model for ceramic-coated metal and consists of two time constants. R_S_ is the solution resistance between the reference electrode and the working electrode, and R_coat_ is the resistance due to the conduction flow of ions through the coating layer, connected in parallel with the constant phase element of the coating layer (CPE_coat_). R_polar_, the polarization resistance at the coating layer–metal interface, is connected in parallel with the CPE_dl_ of the double layer at the coating layer–metal interface. In this model, CPE represents admittance using a power fraction with nonideal behavior in systems with complex dispersion. The arc in the impedance spectra as a depressed semicircle is due to capacitance dispersion. It is appropriate to model with a CPE instead of an ideal capacitor. The admittance of a CPE can be expressed as [40]:Y_CPE_(ω) = 1/Z_CPE_(10)

The fitting data using the equivalent circuit shown in Figure 10c are generally in agreement with the measured data, and the fitting valuesof the equivalent circuit components are shown in Table 3. As the amount of CNT added to the electrolytic solution increases, the value of R_coat_ and R_polar_ increase. When the CNT additive is added, porosity decreases and microcracks become finer in the PEO-coating layer. Furthermore, the presence of CNT in the corroding channel interferes with the conduction flow of ions, resulting in increased R_coat_. In addition, the CNT suppresses penetration of the corrosive ions into the magnesium substrate, so that the R_polar_ is increased.

### 3.5. Heat-Dissipation Property of PEO Coatings

The heat flux of uncoated AZ31 and PEO coatings formed on the AZ31 Mg alloy specimens at different concentrations of CNT is shown in Figure 11. The heat flux of uncoated magnesium substrate was measured as 4317 W/m^2^. Higher heat flux than expected could be attributed to increasing emissivity by surface oxidation by the temperature of the setup and roughness due to polishing. The heat flux of the PEO-coating layer increased to 4700, 4748, 4924, and 5188 W/m^2^ as the addition amount of CNT increased to 0, 2.5, 5, and 10 g/L, respectively. This increase in the heat flux is presumably due to the increase in the thermal conductivity by the CNT with high thermal conductivity in the PEO-coating layer, the decrease of the thickness of the PEO ceramic coatings that interfere with the thermal conduction, and the increase in the emissivity by CNT with high emissivity located on the coating surface.

Figure 12 shows the emissivity of AZ31 Mg alloys and PEO films with and without CNT additives in the electrolyte. The AZ31 magnesium substrate had a low emissivity value of 0.298 in 5–20 μm due to poor photon-emission efficiency from the surface. An emissivity higher than the reported emissivity of ~0.13 appeared to occur due to surface oxidation and roughness from polishing [41,42]. In contrast, the emissivity of the PEO-coated sample without CNT was 0.818. Despite significantly higher emissivity compared to the emissivity of the substrate, the thermal-radiation performance of the PEO film without CNT is hampered by high reflectivity. The emissivity of the CNT-added PEO film was measured to be 0.867. The PEO film with 10 g/L CNT additive, and a dark surface from incorporating a large amount of CNT, showed the highest emissivity due to the effect of high-emissivity CNT. In particular, the addition of CNT significantly increased emissivity in the range of 5–8 μm. The emissivity-enhancement effect of CNT in this study is similar to the effects reported in the literature [43].

## 4. Conclusions

This study investigated the influence of different concentrations of CNT in the PEO coating of an AZ31 Mg alloy on corrosion-resistance and thermal-radiation properties. The CNT additive was incorporated into the oxide film by electrophoresis and agitation during the PEO process. The addition of high-conductivity CNT increased the electrical conductivity of the electrolyte and reduced breakdown voltage. A small-sized microarc occurred during the PEO process, resulting in a dense PEO coating. The thin and dense PEO coating with a large amount of CNT additive exhibited surpassing corrosion resistance in potentiodynamic and EIS experiments. The property of high heat dissipation was obtained by the incorporation of CNT with high thermal conductivity. In addition, high thermal conductivity can be expected with a thin and dense film structure.

## Figures and Tables

**Figure 1 materials-11-02438-f001:**
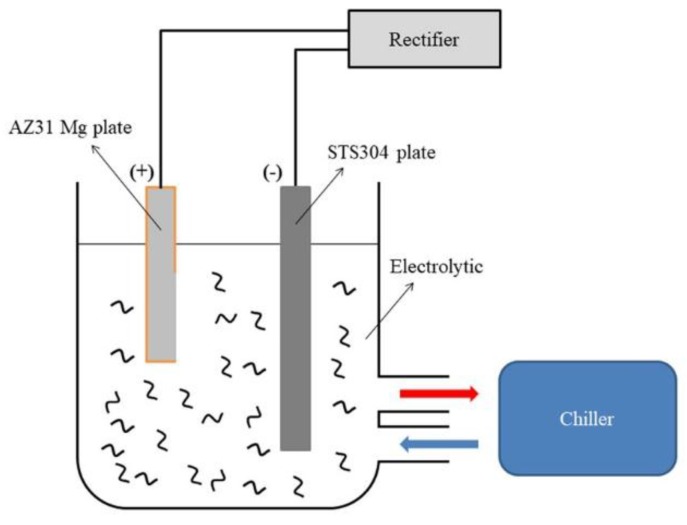
Schematic diagram of the experimental setup used for plasma electrolytic oxidation.

**Figure 2 materials-11-02438-f002:**
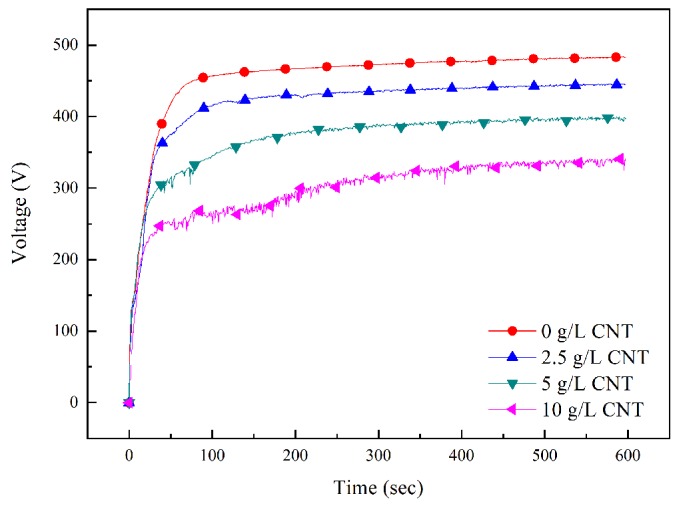
Voltage–time response during the PEO process at different concentration of carbon nanotube (CNT).

**Figure 3 materials-11-02438-f003:**

Optical images of the PEO coatings formed in the electrolytes containing different amounts of CNT: (**A**) 0 g/L CNT; (**B**) 2.5 g/L CNT; (**C**) 5 g/L CNT and (**D**) 10 g/L CNT.

**Figure 4 materials-11-02438-f004:**
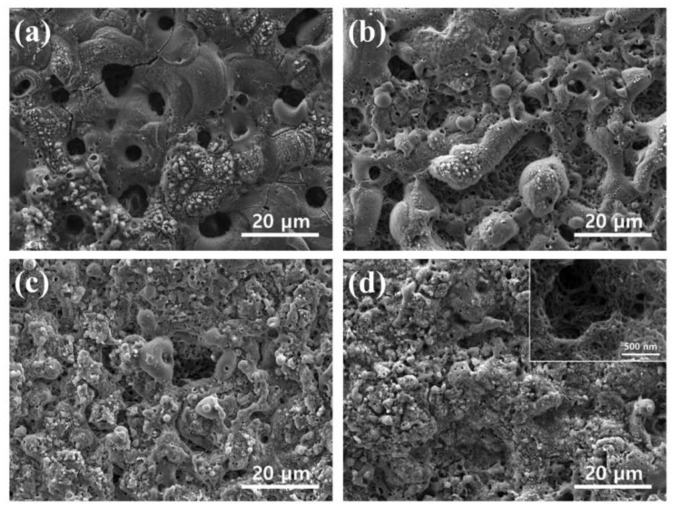
Scanning electron microscopy (SEM) micrographs of the surface morphologies of PEO coatings in different concentration of CNT: (**a**) 0 g/L CNT; (**b**) 2.5 g/L CNT; (**c**) 5 g/L CNT; and (**d**) 10 g/L CNT; the inset is the SEM image of the incorporated CNT.

**Figure 5 materials-11-02438-f005:**
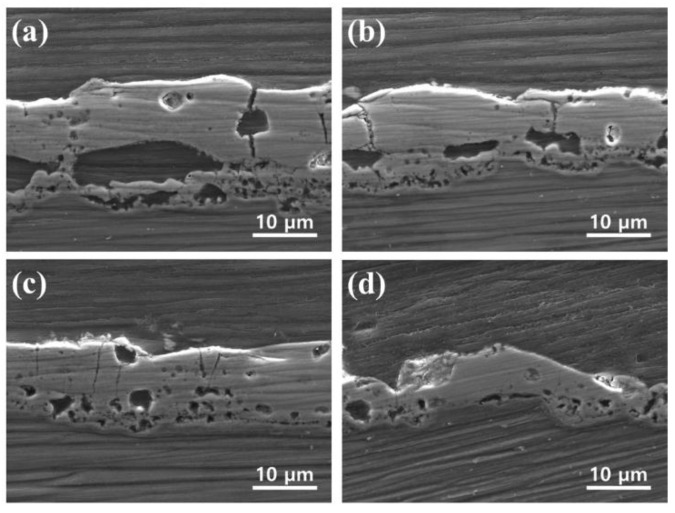
Cross-sectional SEM images of PEOcoated specimens formed in the electrolytes containing different amounts of CNT: (**a**) 0 g/L CNT; (**b**) 2.5 g/L CNT; (**c**) 5 g/L CNT; and (**d**) 10 g/L CNT.

**Figure 6 materials-11-02438-f006:**
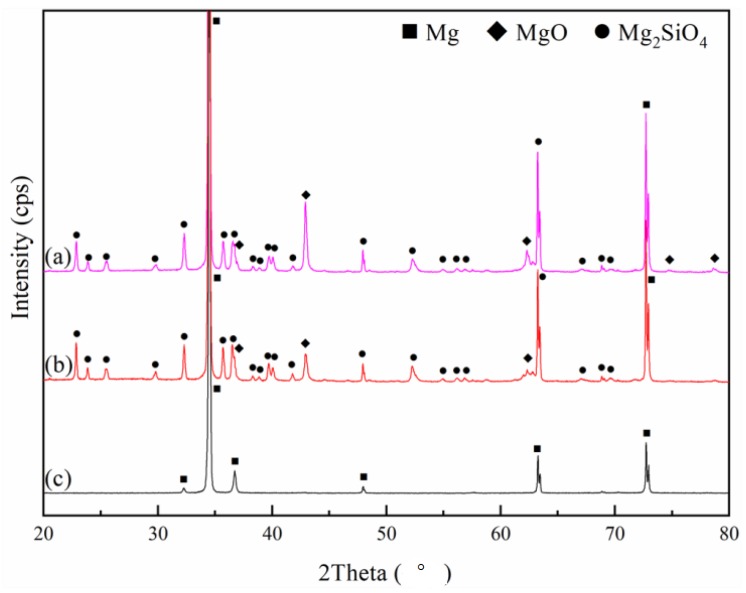
X-ray diffraction (XRD) patterns of PEO coatings and substrate: (**a**) 10 g/L CNT PEO coating; (**a**) 0 g/L CNT PEO-coating; and (**c**) AZ31 Mg substrate.

**Figure 7 materials-11-02438-f007:**
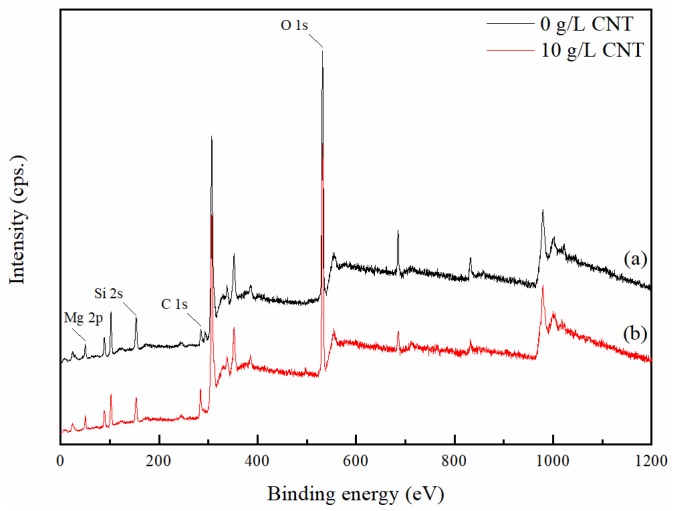
X-ray photoelectron spectroscopy (XPS) spectra of the PEO coatings in an electrolyte containing different concentration of CNT: (**a**) 0 g/L CNT; (**b**) 10 g/L CNT.

**Figure 8 materials-11-02438-f008:**
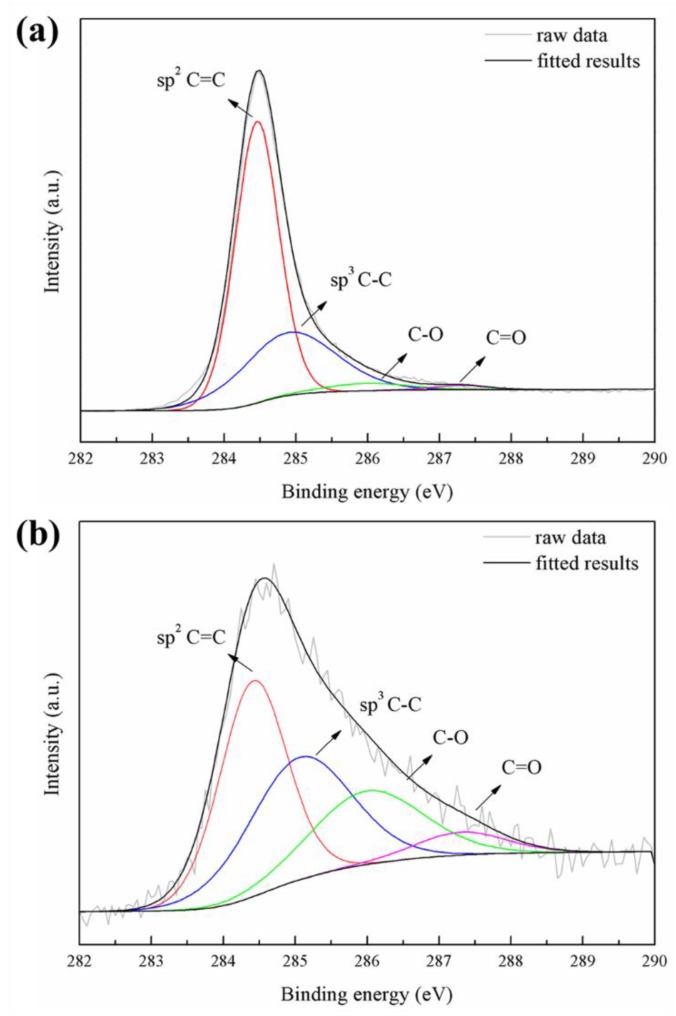
High-resolution XPS spectra of element-detected carbon: (**a**) pristine CNT and (**b**) 10 g/L CNT PEO-coated film.

**Figure 9 materials-11-02438-f009:**
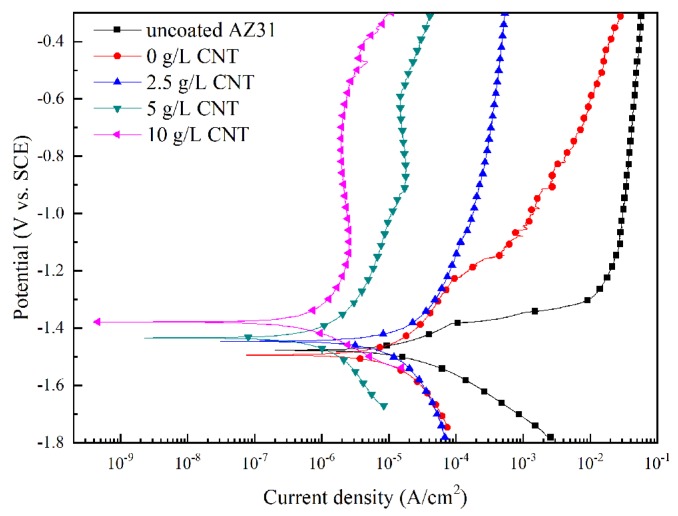
Potentiodynamic polarization curves of the PEO coatings formed in different CNT concentration of electrolytes.

**Figure 10 materials-11-02438-f010:**
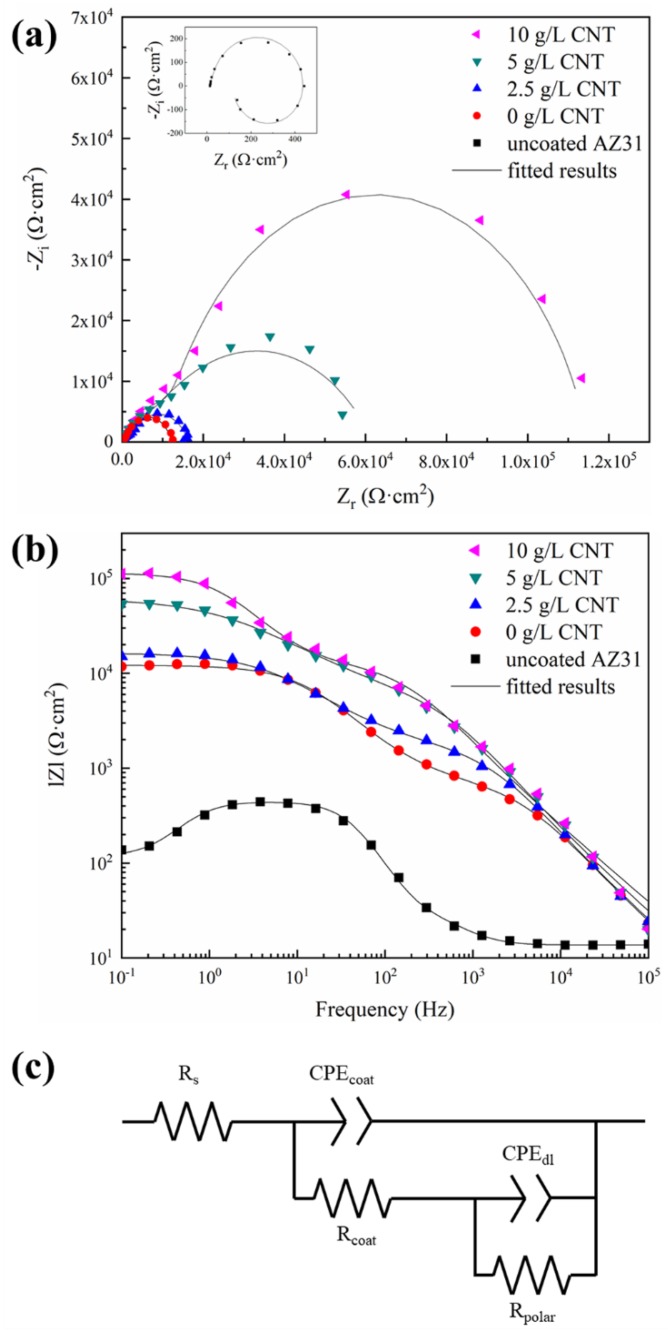
Electrochemical impedance behavior of the PEO coatings in different concentration of CNT: (**a**) Nyquist plot; (**b**) Bode plot; and (**c**) equivalent circuit.

**Figure 11 materials-11-02438-f011:**
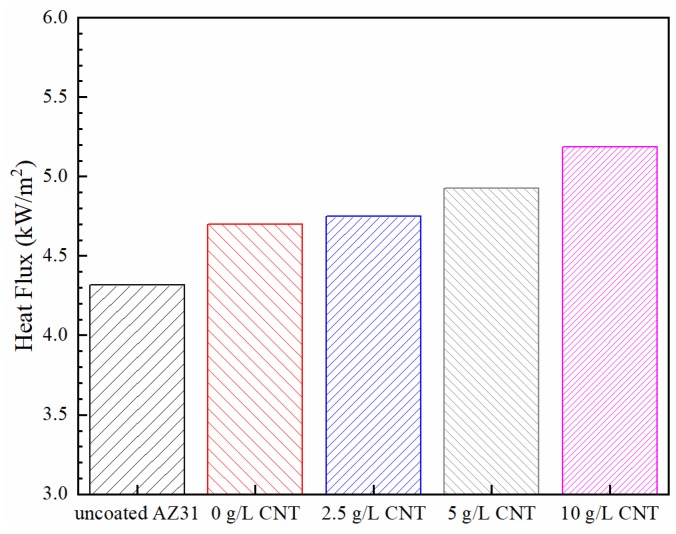
Heat flux of the AZ31 Mg substrate and the PEO coatings in different concentrations of CNT.

**Figure 12 materials-11-02438-f012:**
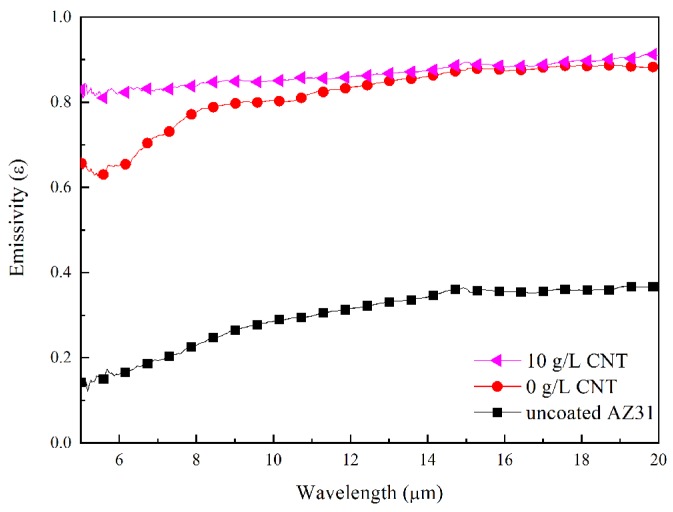
Emissivity of PEO coatings with and without CNT and the AZ31 Mg substrate measured at 100 °C, via Fourier transform infrared spectroscopy (FT-IR). Figure 13 shows the heat-transfer mechanism of the PEO coating with and without CNT additive. The thermal conductivity of MgO acting as an thermal insulator in the coating layer was quite low, at about 30 W⋅m^−1^⋅k^−1^ [44]. However, the composite PEO coating could improve thermal conductivity by incorporating a CNT additive with high thermal conductivity (2000–6000 W⋅m^−1^·k^−1^) into the oxide layer [45]. In addition, the PEO coating with a CNT additive has a thin and dense structure. When the thickness of the coating was thin, the distance of the heat-conduction path having low thermal conductivity was shortened. The size of the pore band also decreased, and the area of the heat-obstructing vacancies was reduced, thereby exhibiting improvement in heat-conductive ability. When the heat is transferred to the coating surface, a high emissivity oxide coating with a CNT additive caused a large amount of thermal radiation, which improved the heat-dissipation capability of the PEO coating.

**Figure 13 materials-11-02438-f013:**
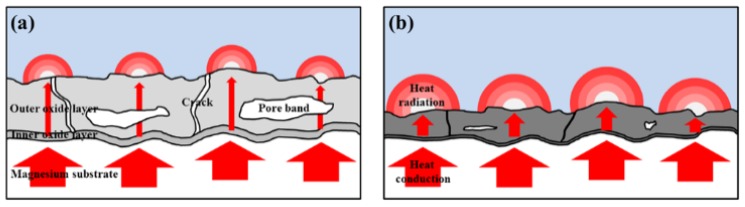
Schematic diagram of the heat-transfer mechanism of the PEO coatings with and without CNT additive: (**a**) the PEO coating without CNT additive and (**b**) the PEO coatings with CNT additive.

**Table 1 materials-11-02438-t001:** Specification of the electrolytes for the plasma electrolytic oxidation (PEO) process in AZ31 Mg alloy.

Sample	CNT (g/L)	KOH (g/L)	KF (g/L)	Na_2_SiO_3_ (g/L)	pH	Conductivity
0 g/L CNT	0	2	2	6	12.3	13.7
2.5 g/L CNT	2.5	2	2	6	12.3	14.8
5 g/L CNT	5	2	2	6	12.3	16.4
10 g/L CNT	10	2	2	6	12.3	18.1

**Table 2 materials-11-02438-t002:** Tafel analysis results of uncoated and PEO-coated samples in an electrolyte containing different concentration of CNT.

Sample	Potential (V)	Current (A/cm^2^)	b_a_ (mV)	b_c_ (mV)
uncoated AZ31	−1.481	1.12 × 10^−5^	94	82
0 g/L CNT	−1.493	7.14 × 10^−6^	128	118
2.5 g/L CNT	−1.445	3.67 × 10^−6^	55	100
5 g/L CNT	−1.473	1.43 × 10^−6^	132	108
10 g/L CNT	−1.376	4.80 × 10^−7^	146	111

**Table 3 materials-11-02438-t003:** Equivalent circuit parameters from fitted Electrochemical Impedance Spectroscopy (EIS) results of PEO coatings formed in the electrolytes containing different amounts of CNT.

Sample	R_coat_ (Ω⋅cm^2^)	CPE_coat_ (Ω⋅^−1^⋅s^n^⋅cm^−2^)	n_coat_	R_polar_ (Ω⋅cm^2^)	CPE_dl_ (Ω⋅^−1^⋅s^n^⋅cm^−2^)	n_dl_
0 g/L CNT	9.47 × 10^2^	6.37 × 10^−8^	0.79	1.13 × 10^4^	2.99 × 10^−9^	0.78
2.5 g/L CNT	2.09 × 10^3^	5.25 × 10^−8^	0.80	1.43 × 10^4^	3.23 × 10^−9^	0.69
5 g/L CNT	7.60 × 10^3^	3.60 × 10^−8^	0.81	5.43 × 10^5^	1.04 × 10^−9^	0.60
10 g/L CNT	1.60 × 10^4^	3.97 × 10^−8^	0.78	9.87 × 10^5^	3.21 × 10^−9^	0.86

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
