# Peer review of "Effects of a Carbon Nanotube Additive on the Corrosion-Resistance and Heat-Dissipation Properties of Plasma Electrolytic Oxidation on AZ31 Magnesium Alloy"

_materials, 2018, doi:10.3390/ma11122438_

Reviewer 1 Report

  Minor corrections are needed. The authors should note that plural of "radius" is. Color markings in the pdf text should be helpful to provide corrections.

Author Response

November, 27, 2018

Dear Editor

I sincerely appreciate your hard work! I did my best to write responses to your points. All changes can be tracked with change tracking.

Sincerely,

Wonsub Chung 

Reviewer 2 Report

1.      There is no information in the Introduction concerning the real practical use of oxide coatings as heat dissipators.

2.      It is not clear why only the radiation mechanism is concerned. What is the intended area of practical application of the coatings?

3.      Ceramics are thermal insulators with low coefficient of thermal conductivity and there is always temperature difference between metal and surface of the coating. Discussion and calculation of thermal conductivity of a system metal/oxide coating/working media should be provided.

4.      It is stated that: “excellent emissivity and thermal conductivity was incorporated into the oxide film to produce coatings with excellent heat dissipation capabilities”, but there is no comparison of obtained results with other researches. The obtained values are quite average and typical. The statement should be supported with discussion. Starting from 11 microns to 20 the obtained values of emissivity for PEO and PEO+CNT are the same within the error. Disclose, please, potential benefits for the observed difference in emissivity in the range 5-8 microns taking into account that this range is more characteristic for the bodies with the temperature of several hundred degrees. Is it applicable for Mg?

5.      It is stated that: “CNT present in pores and microcracks is likely to inhibit the evolution of corrosive ions” and “These results demonstrate that the incorporation of CNT additive into the oxide film improve the corrosion resistance properties of PEO coating”, “the presence of CNT in the corroding channel interferes with the conduction flow of ions, resulting in increased Rcoat. In addition, the CNT suppresses penetration of the corrosive ions into the magnesium substrate, so that the Rpolar is increased”. Disclose, please, the mechanism of protection provided by CNTs taking into account that barrier film should be dielectric or semiconductive and CNTs have electronic conductivity, reducing resistivity of the coating. And how CNTs increase R.

6.      “current density (-1.481 V) and low corrosion potential (1.12 x 10-5 A/cm2)”

7.      “decrease of about two orders of magnitude relative to the uncoated AZ31 Mg alloy” 1.12*10^-5/4.8*10^-7=23.3  Not two orders. Correct, please.

Author Response

November, 27, 2018

Dear Editor

I sincerely appreciate your hard work! I did my best to write responses to your points. All changes can be tracked with change tracking. Please see the attached PDF file.

Sincerely,

Wonsub Chung 

Round  2

Reviewer 2 Report

OK